# NF2-Related Schwannomatosis (NF2): Molecular Insights and Therapeutic Avenues

**DOI:** 10.3390/ijms25126558

**Published:** 2024-06-14

**Authors:** Bae-Hoon Kim, Yeon-Ho Chung, Tae-Gyun Woo, So-mi Kang, Soyoung Park, Minju Kim, Bum-Joon Park

**Affiliations:** 1Rare Disease R&D Center, PRG S&T Co., Ltd., Busan 46274, Republic of Korea; bk728@prgst.com (B.-H.K.);; 2Department of Molecular Biology, College of Natural Science, Pusan National University, Busan 46241, Republic of Korea

**Keywords:** NF2, NF2-related schwannomatosis, Merlin, YAP, Yes-associated protein 1, RKIP, Raf kinase inhibitory protein, PPI, protein–protein interaction

## Abstract

NF2-related schwannomatosis (NF2) is a genetic syndrome characterized by the growth of benign tumors in the nervous system, particularly bilateral vestibular schwannomas, meningiomas, and ependymomas. This review consolidates the current knowledge on NF2 syndrome, emphasizing the molecular pathology associated with the mutations in the gene of the same name, the *NF2* gene, and the subsequent dysfunction of its product, the Merlin protein. Merlin, a tumor suppressor, integrates multiple signaling pathways that regulate cell contact, proliferation, and motility, thereby influencing tumor growth. The loss of Merlin disrupts these pathways, leading to tumorigenesis. We discuss the roles of another two proteins potentially associated with *NF2* deficiency as well as Merlin: Yes-associated protein 1 (YAP), which may promote tumor growth, and Raf kinase inhibitory protein (RKIP), which appears to suppress tumor development. Additionally, this review discusses the efficacy of various treatments, such as molecular therapies that target specific pathways or inhibit neomorphic protein–protein interaction caused by *NF2* deficiency. This overview not only expands on the fundamental understanding of NF2 pathophysiology but also explores the potential of novel therapeutic targets that affect the clinical approach to NF2 syndrome.

## 1. Introduction

NF2-related schwannomatosis (NF2) syndrome is a rare genetic disorder primarily characterized by the development of benign tumors in the central nervous system, including bilateral vestibular schwannomas, brain tumors, spinal tumors, and peripheral nerve tumors. Adults typically present with hearing loss and balance disturbance, and children with ocular, dermatological, and neurological signs [1,2]. NF2 syndrome is caused by germline or mosaic pathogenic variants in the *NF2* gene located in the long arm of chromosome number 22 (22q12.2). The estimated prevalence of NF2-related schwannomatosis is 1:50,000 with a birth incidence of 1:28,000 [3,4]. NF2 is passed down in an autosomal dominant fashion. Around half of the individuals diagnosed with NF2 have inherited it from a parent who also has the condition. The other half develop NF2 due to a new (de novo) mutation in the *NF2* gene. Between 25% and 50% of those with a new *NF2* mutation exhibit somatic mosaicism for the mutation. If a parent is shown to have mosaic NF2, it rules out the possibility that they have the inherited form of the disorder. Each child of a person with NF2 has a potential 50% chance of inheriting the mutation: children of someone with a germline mutation have a 50% risk, while children of someone with mosaic NF2 have a lower than 50% risk. Once the NF2 mutation is identified in a family, genetic testing can be performed before birth or during the early stages of pregnancy [5]. The merlin protein encoded by the *NF2* gene is a membrane-cytoskeleton scaffolding protein, i.e., linking actin filaments to cell membrane or membrane glycoproteins [6]. Human merlin is predominantly found in the Schwann cells of nervous tissue, but also in several other fetal tissues, and is mainly located in adherens junctions [7,8,9,10]. Merlin is a FERM (Four point one, Ezrin, Radixin, Moesin) domain-containing protein whose loss results in defective morphogenesis and tumorigenesis in multiple tissues. The loss of merlin leads to altered cell adhesion, increased cell migration, and reduced apoptosis rates, contributing to tumor formation. It is known that overall disease severity is strongly associated with the type and extent of mutation of the *NF2* gene. Truncating mutations are associated with the most severe phenotype, earlier presentation, and severe morbidity [11,12,13,14,15]. Clinical diagnosis is confirmed by neuroimaging and genetic testing. Inactivating germline mutations in NF2 patients and associated tumors suggest a critical role in tumorigenesis. NF2 syndrome also involves peripheral nerve pathology, which can cause high morbidity including pain, motor, and sensory loss. Spinal tumors in NF2 syndrome include both intramedullary and extramedullary spinal tumors. Up to now, there is no surrogate marker for the clinical efficacy assessment except for magnetic resonance imaging (MRI). Therefore, MRI is crucial in detecting these lesions and in assessing the efficacy of the treatment [16,17].

Up to now, it is evident that NF2 syndrome is a monogenic disease caused only by the mutation of the *NF2* gene encoding the Merlin protein, not by any other genes. When the homeostatic condition is interrupted from the absence of the functional Merlin protein in an *NF2* deficient cell, neomorphic protein–protein interactions (neo-PPIs) can be born in the affected cells for survival in the new condition. PPIs are the main biochemical mechanisms of cellular life, and are frequently disturbed in disease conditions. PPIs create many kinds of functional multi-protein complexes and give roles in basic processes such as protein transport, transcription, translation, interaction/communication between cells, protein modification, signaling cascades, and functional holoenzyme to each complex. The complex network of direct interactions between proteins, known as the interactome, has been widely recognized as an important concept for therapeutic targets in various diseases [18,19].

This review summarizes (i) the current knowledge on NF2 syndrome, emphasizing the molecular pathology associated with the *NF2* gene mutations and the subsequent dysfunction of its product, the Merlin protein; (ii) discusses the roles of two proteins potentially associated with NF2 deficiency: Yes-associated protein 1 (YAP) and Raf kinase inhibitory protein (RKIP); and (iii) introduces recent advances in targeted therapies and ongoing clinical trials for managing NF2 syndrome.

## 2. History of NF2 Syndrome

The most common tumors associated with NF2 syndrome are vestibular schwannomas, also known as acoustic neuromas, which develop on the nerves leading from the inner ear to the brain. Other nervous system tumors can also occur. The history and definition of NF2 syndrome have evolved over time. Neurofibromatosis was first described in the late 19th century. Initially, there was no clear distinction between what would later be known as NF1 and NF2 syndrome [14]. This period was marked by an increasing understanding of the clinical features of neurofibromatosis. For example, Ruggieri et al. explore the early history of NF2 syndrome and related forms, discussing the earliest descriptions of acoustic neuron tumors [20]. In the mid-20th century, the distinction between NF1 (previously known as von Recklinghausen’s disease) and NF2 became clearer. NF1 and NF2 syndrome were recognized as separate conditions due to their different clinical presentations and genetic causes. NF1 is characterized by multiple café-au-lait spots and neurofibromas on the skin, while NF2 syndrome is tightly associated with bilateral vestibular schwannomas [21]. The location on 22q. of the *NF2* gene was suggested in 1986, and it was identified in 1993. This gene, located on chromosome 22, is responsible for the production of a protein called merlin (or schwannomin), which suppresses tumors. Mutations in the *NF2* gene lead to the development of the various tumors associated with the disorder. Evans (1999) provides an overview of the genetic and clinical features of NF2 syndrome, including the discovery of the *NF2* gene [22]. With the identification of the *NF2* gene, genetic testing became a tool for diagnosing NF2 syndrome. Advancements in imaging techniques, like MRI, also improved the diagnosis process. Treatment options have expanded to include surgery, radiation therapy, and, more recently, targeted drug therapies. Tysome et al. (2014) discuss current concepts in the management of vestibular schwannomas in NF2 syndrome. Research continues, aiming at better understanding the *NF2* gene and its functions, and developing effective treatments [23]. This includes exploring gene therapy and newer targeted therapies, and improving surgical and radiation techniques [24,25,26]. At present, there is no cure for NF2 syndrome, but treatments are available to manage symptoms and complications associated with the condition. The treatment approach for NF2 syndrome typically includes the surgical removal of tumors, which is often recommended, especially for those tumors that cause symptoms or have a risk of causing further complications [27]. However, surgery can carry risks, such as damage to surrounding nerves. Stereotactic radiosurgery (such as Gamma Knife radiosurgery) can be used to target and shrink tumors, particularly when surgery is not an option [14]. For patients experiencing hearing loss, hearing aids or cochlear implants may improve hearing and quality of life. While there is no medication that cures NF2 syndrome, some drugs may help reduce the growth of certain tumors [28]. Research into NF2 syndrome is ongoing, and new treatments, including gene therapy and targeted molecular therapies, are being explored. These treatments aim to target the underlying genetic mutations or pathways that contribute to tumor growth [29].

## 3. Merlin Protein as a Tumor Suppressor

NF2 syndrome’s molecular pathology is closely associated with mutations in the *NF2* gene and the ensuing dysfunction of the merlin protein, contributing to the emergence of multiple nervous system tumors. The tumorigenesis in NF2 syndrome can be explained by the tumor suppressor function of the Merlin protein, which is produced by the *NF2* gene. Merlin serves as a critical tumor suppressor across various pathways and is recognized as a negative regulator of growth in fruit flies [30,31]. It controls cell proliferation and apoptosis in the developing imaginal discs of Drosophila, positioning it upstream of the Hippo signaling cascade [32].

Merlin has been identified as a negative regulator of Rac signaling. The overexpression of merlin suppressed the Rac-induced activation of c-Jun N-terminal kinase (JNK) and the activation of the AP-1 transcriptional activator. Conversely, in *Nf2*-deficient fibroblasts, basal JNK activity was observed to be elevated, as was the activity of AP-1 [33]. Merlin interacted with p21-activated kinase 1 (Pak1), a crucial mediator of Rac/cdc42 signaling, and the activity of the kinase was diminished by the interaction, suggesting that the absence of merlin expression led to the inappropriate activation of Pak1 [34]. Merlin orchestrates adherens junction (AJ) stabilization and the negative regulation of epidermal growth factor receptor (EGFR) signaling, confining EGFR in a compartment from which it cannot signal or be internalized. This phenomenon was interpreted through the NHE-RF1–mediated, contact-dependent association of Merlin and EGFR. It provides a new mechanistic insight regarding the contact-dependent inhibition of proliferation and a therapeutic strategy for *NF2* gene mutant tumors [35]. Furthermore, it has been demonstrated that Merlin is crucial for the establishment of stable AJs in epidermal keratinocytes and plays a critical role in AJ establishment in vivo by directly associating with α-catenin and linking it to Par3 [36]. Merlin’s primary tumor suppressive functions in response to adhesive signaling are also attributed to its management of oncogenic gene expression through the regulation of Hippo signaling. This regulation occurs through the suppression of the CRL4DCAF1 E3 ubiquitin ligase by merlin, which curtails the inhibition of the Lats kinases [37]. It has been revealed that tight-junction-associated proteins including Merlin, Angiomotin, Patj, and Pals1 form a complex, showing Angiomotin acting downstream of Merlin and upstream of Rich1 in the regulation of Rac1 and Ras-MAPK pathways. This effectively results in the reduction of Rac1 and Ras-MAPK pathways [38].

Interestingly, it has been found that the Sumoylation of Merlin is essential for its tumor-suppressive activity. This post-translational modification at the Lysine residue (K76) facilitates Merlin’s intramolecular and intermolecular binding activities and influences its distribution between the cytoplasm and nucleus [39]. Previously, Merlin’s tumor suppressor function was shown to be activated by dephosphorylation at serine 518, a process facilitated by the myosin phosphatase MYPT-1–PP1δ. Blocking this phosphatase leads to a loss of Merlin function [40]. However, recent findings propose that its tumor suppressor function is independent of Serine 518 phosphorylation, challenging the prevailing view that phosphorylation governs Merlin’s structure and activity [41]. Merlin interacts with both Ras and p120RasGAP, yet it does not enhance the catalytic activity of RasGAP. Hence, the interactions with Ras and RasGAP could fine-tune Ras signaling, further contributing to Merlin’s tumor suppressor activity and potentially implicating Merlin in the regulation of other small GTPases [42].

Recently, research has shown that Merlin interacts with TGFβ receptor 2 (TβR2) and maintains TβR2 expression. The absence of Merlin decreases TβR2 levels and triggers non-canonical TGFβ signaling through TβR1 kinase activity, leading to the phosphorylation and subsequent degradation of Raf kinase inhibitor protein (RKIP) [24]. Moreover, under *NF2*-deficient conditions, a selective chemical, Nf18001, that blocks the interaction between TβR1 and RKIP has been shown to inhibit tumor growth and encourage the maturation of schwannoma cells into mature Schwann cells [25]. These studies may provide a possible mechanism for the impact of Merlin loss in the tumorigenesis of NF2 syndrome patients. Understanding these molecular mechanisms is critical for the diagnosis and management of NF2 syndrome.

## 4. YAP Protein as a Tumor-Inducing Factor

The molecular pathology of NF2 syndrome, concerning the YAP protein, has been thoroughly investigated by researchers. As a modulator of the Hippo signaling pathway, merlin activates LATS1/2 kinases and subsequently inhibits YAP/TAZ. Below are significant findings that delineate the association and influence of Yes-associated protein (YAP) in *NF2*-deficient conditions.

In Drosophila, merlin governs cell proliferation and apoptosis by mediating signals through the Hippo pathway to curtail the function of the transcriptional coactivator Yorkie. This pathway is preserved in mammals [32,43]. Based on these insights, human meningioma cell lines were utilized to assess growth regulation by merlin and explore the connection between Merlin status and YAP, the mammalian equivalent of Yorkie. Merlin deficiency in meningioma cells leads to enhanced YAP expression and nuclear presence. Reducing YAP levels in merlin-deficient meningioma cells can alleviate the effects of merlin deficiency on cellular proliferation, suggesting that merlin regulates cancer cell growth through the suppression of YAP [44]. It was proposed that the merlin protein constrained the proliferation of neural progenitor cells (NPCs) in the mammalian dorsal telencephalon. Merlin was situated at the apical region of NPCs, and the lack of merlin resulted in the pronounced malformation of the hippocampus in adult mice. Merlin curbed the proliferation of neural progenitor cells by blocking YAP/TAZ, vital for brain development control, possibly via a pathway distinct from the standard Hippo pathway. *Nf2* mutants exhibited an excessive growth of cortical hem and hippocampal primordium due to heightened YAP/TAZ activity, highlighting a crucial developmental role. The excessive expression of human YAP in neural progenitor cells (NPCs) resulted in hippocampal deformities similar to those observed in *Nf2* mutants. Crucially, deleting Yap in the context of *Nf2* mutations significantly improved hippocampal development [45]. In *NF2* gene-positive meningiomas, areas with diminished merlin levels exhibited more cells with nuclear YAP. Cell density and extracellular matrix rigidity further regulated YAP localization, indicating that YAP activation in meningiomas is affected by merlin, cell density, and matrix rigidity [46]. *NF2* gene loss in thyroid cancer facilitates YAP transcriptional activity, augmenting RAS expression. This illustrates that YAP can modulate gene expression independently of Merlin’s conventional role in the Hippo pathway, associating *NF2* gene loss with RAS-driven cancers [47]. When the expression of YAP/TAZ was suppressed in *NF2*-deficient cells, established *NF2*-deficient tumors regressed. Depleting YAP/TAZ reduces glycolysis-dependent growth and boosts mitochondrial respiration and reactive oxygen species (ROS) accumulation, signifying a substantial role in NF2-related tumorigenesis [48]. In the absence of the *NF2* gene, Motins significantly amass to limit the full activation of YAP/TAZ and avert rapid tumorigenesis. Consequently, *NF2* gene deficiency not only activates YAP/TAZ by blocking LATS1/2 but also solidifies Motins to restrain YAP/TAZ activity. The elevation of Motins upon the deletion of the *NF2* gene acts as a tactic to prevent unchecked disturbances of the Hippo signaling and may contribute to the benign nature of most *NF2*-mutated tumors [49].

In summary, YAP plays a pivotal role in the pathology of NF2 syndrome, especially in tumor growth and development. The dynamic between Merlin and YAP is essential in regulating cell proliferation and survival, with implications for targeted therapies in NF2-associated tumors.

## 5. RKIP as a Tumor Suppressor

From 1990 to 2010, numerous investigations explored the role of Raf Kinase Inhibitory Protein (RKIP) as a tumor suppressor in various cancers, focusing on its role in curbing metastasis signaling and its effects on cancer cell growth and migration. These investigations underscored RKIP’s complex role in halting cancer progression through diverse mechanisms, including the adjustment of signaling pathways, the oversight of microRNAs, the curtailment of cell growth and metastasis, and its effects on the tumor microenvironment. The outcomes underscore RKIP’s potential as a therapeutic target and its prognostic value in cancer [50,51,52]. Here are the principal findings about RKIP from the research. The diminished or absent expression of RKIP aligns with breast cancer metastasis, positioning RKIP as a critical metastasis suppressor gene that must be downregulated for metastases to form in human breast cancer [53]. This confirms findings from cell culture and animal models, indicating that in human breast cancer, RKIP serves as a metastasis suppressor gene whose downregulation is necessary for metastasis formation. RKIP’s expression occurs independently of other indicators of breast cancer progression and prognosis [53]. RKIP shapes metastasis suppressor pathways in breast cancer, orchestrating genes like HMGA2 and BACH1, and affecting metastasis-related genes [54]. In ovarian cancer, RKIP levels inversely relate to cell invasiveness, and its heightened expression deters cell growth, migration, and anchorage-independent growth, signifying its role as a metastasis suppressor gene in epithelial ovarian cancer [55]. Similarly, RKIP restricts the MAP kinase (MAPK), G protein-coupled receptor kinase-2, and NF-κB signaling cascades, serving as a metastasis suppressor in prostate cancer and breast cancer by deterring invasion and metastasis through a signaling sequence involving MAPK, Myc, LIN28, let-7, and HMGA2 [56]. RKIP also restricts cancer cell invasion through extracellular matrix barriers by managing the expression of matrix metalloproteinases (MMPs), particularly MMP-1 and MMP-2, providing insights into its role as a metastasis suppressor by negatively regulating MMP expression in breast cancer, colon cancer, and melanoma [57]. Recently, RKIP has been shown to modulate the tumor microenvironment, particularly by controlling the infiltration of specific immune cells and the secretion of pro-metastatic factors. This reinforces RKIP’s role not only in direct tumor cell behavior but also in influencing the surrounding environment, which is vital for cancer progression and metastasis in breast cancer cells [58]. The downregulation of RKIP expression is a significant factor in the activation of the IGF-I/ERK/MAPK pathway during human hepatocarcinogenesis [59]. This report showed that RKIP expression is reduced in human HCC compared with adjacent peritumoral tissues. Low RKIP levels were linked with increased extracellular signal-regulated-kinase (ERK)/MAPK pathway activation. Reconstitution experiments countered IGF-I-mediated MAPK pathway activation, leading to a reduced nuclear accumulation of phospho-ERK. Conversely, the suppression of RKIP expression using small interfering RNA initiated the activation of the ERK/MAPK pathway. The ectopic expression of RKIP altered HCC cell growth and migration, pointing to its role in human hepatocarcinogenesis. Intriguingly, Kim et al. discovered that RKIP downregulation acts as a mechanism of sorafenib resistance in hepatocellular carcinoma cell lines, underscoring the significance of RKIP in therapeutic resistance [60]. Additionally, Zhang Lin et al. demonstrated that RKIP functions as a tumor suppressor by influencing the biological characteristics of hepatocellular carcinoma cells, suggesting potential therapeutic applications [61]. A study revealed that miR-224 expression is elevated in breast cancer cell lines, particularly in highly invasive cells, and directly represses RKIP gene expression. This suggests that miR-224 plays a crucial role in the metastasis of human breast cancer cells to the bone by directly inhibiting the tumor suppressor RKIP [62]. The overexpression of RKIP was shown to deter cell growth and invasion in breast cancer cell lines by elevating miR-185, which targets and restrains HMGA2, a gene implicated in cancer metastasis. This study highlights the potential therapeutic value of signaling pathway profiles including RKIP and miR-185 [63]. Research has shown that RKIP expression is markedly reduced in pancreatic carcinoma tissues and is associated with patient outcomes. The overexpression of RKIP in pancreatic adenocarcinoma cells curbed cell proliferation and metastasis, and downregulated the Raf-1-MEK1/2-ERK1/2 signaling pathway, indicating its potential as a therapeutic target for anti-metastatic strategies in pancreatic cancer [64].

Recently, Park and his colleagues have studied the relationship between Merlin and RKIP, suggesting that there are specific interactions that may influence tumor behavior and therapeutic responses: They found that Merlin blocks Snail-mediated p53 suppression and it is stabilized by RKIP, suggesting a cooperative role in blocking p53 inhibition in mesothelioma [65]. And then, they reported that the loss of the *NF2* gene induces non-canonical and oncogenic TGFβ signaling mediated by TGFβ receptor1, leading to the phosphorylation and degradation of RKIP. This mechanism results in MAPK activation and oncogenic signaling in *NF2*-deficient cells. Furthermore, it was suggested that blocking the kinase activity of TGFβ receptor 1 could potentially restore cell differentiation and suppress growth in NF2-related conditions [24]. Finally, they highlighted that a selective RKIP inducer (a novel chemical called Nf18001), which does not block the kinase activity of TGFβ receptor 1, can inhibit tumor growth and promote schwannoma cell differentiation under *NF2*-deficient conditions. This report involves reducing SOX2 expression and increasing SOX10 expression through Nf18001 [25]. These findings indicate a protein–protein interaction (PPI) between Merlin and RKIP, primarily influencing tumor growth, signaling pathways, and gene regulation, which could inform future therapeutic strategies targeting these pathways.

## 6. Pre-Clinical Research on Possible Treatments of NF2 Syndrome

Numerous attempts have been made to effectively develop treatment strategies for NF2 syndrome using mouse models and cell lines derived from human patients. Beyond the experimental investigations into the treatment of the syndrome, multiple clinical trials have been implemented. Predicated on observations that group I p21-activated kinases (Paks) attach to and are inhibited by the *NF2* gene-encoded protein Merlin, the signaling and anti-tumor properties of three group-I specific Pak inhibitors were evaluated in *NF2*−/− meningiomas both in vitro and within an orthotopic mouse model. It was observed that these Pak inhibitors curtailed the proliferation and motility of both benign (Ben-Men1) and malignant (KT21-MG1) meningioma cells, indicating that the agents may prove beneficial in managing *NF2*-deficient meningiomas [66,67].

Following the discovery that diminishing RKIP levels leads to Merlin instability and tumorigenic advancement in mesothelioma [65], Park and his colleagues established that Merlin deficiency lessens the stability of transforming growth factor-beta receptor 2 (TβR2), and physical forces such as pressure or dense materials facilitate unwarranted protein–protein interactions (PPI) between TbR1 and RKIP. This new interaction results in the phosphorylation and degradation of RKIP. The decline of RKIP subsequently facilitates MAPK activation as well as Snail-mediated p53 inhibition in *NF2*-deficient cells. Subsequently, they discovered that TGFβ receptor 1 kinase inhibitor TEW7197 could diminish tumor formation in NF2 syndrome-model mice and impede growth in *NF2*-deficient schwannoma and MEF cell lines [24,65]. This implied that targeting TbR1 for treatment could be promising, yet the potential side effects from inhibiting TGFb signaling through TbR1 remained a concern. To source drugs that alleviate this emerging NF2 syndrome pathology without disturbing canonical TGFβ signaling, they screened candidate compounds that selectively target the new PPI between TbR1 and RKIP, leading to a reduction in RKIP. A new chemical, Nf18001, identified from the screening, curbed proliferation and fostered the differentiation of HEI-193 human schwannoma cells, and effectively inhibited tumor growth in an allograft model, while the SMAD cascades from TGFβ signaling remained unaffected [25].

Based on evidence that Merlin inhibits tumorigenesis from within the nucleus via a Merlin-CRL4DCAF1-LATS1/2-YAP signaling axis, it was proposed that this signaling could offer valuable therapeutic targets for treating *NF2* gene-inactivated cancers [37,68]. In a recent study, Cooper et al. endeavored to ascertain whether CRL4DCAF1 could serve as an effective therapeutic target for countering *NF2* gene loss-related tumorigenesis. In their study, the authors employed a small molecule drug (MLN4924) to indirectly impede CRL4DCAF1 ubiquitin ligase activity by targeting its upstream activator NEDD8-activating enzyme (NAE). Indeed, MLN4924 inhibited CRL4DCAF1 activity and LATS1/2 ubiquitination in vitro.

Cooper et al. established tumor xenografts in immunocompromised mice with *NF2*-mutant cell lines. MLN4924 slowed the rate of tumor growth, though it could not completely halt growth or induce tumor regression. From the finding that MLN4924 treatment had no impact on mTORC1 activity or its downstream signaling through S6 in *NF2*-mutant cells, they speculated that *NF2* gene loss likely activates this pathway independently of CRL4DCAF1. To examine whether mTOR signaling sustains *NF2*-mutated cancer cells during CRL4DCAF1 blockade, Cooper et al. treated mesothelioma tumor xenografts with MLN4924 and GDC-0980 (a dual mTOR/PI3K inhibitor). The combined MLN4924/GDC-0980 treatment completely eradicated tumor growth in vivo for four weeks [69,70].

In another initiative to develop an appropriate treatment for NF2 syndrome, the NF2 consortium identified brigatinib, an FDA-approved inhibitor of multiple tyrosine kinases (TKI) including anaplastic lymphoma kinase (ALK), as a potent suppressor of tumor growth in established *NF2*-deficient xenograft meningiomas and a genetically engineered murine model of spontaneous NF2 schwannomas. They demonstrated that brigatinib inhibited several tyrosine kinases, including EphA2, Fer, and focal adhesion kinase 1 (FAK1), but not ALK, demonstrating that an effective systems biology approach for preclinical therapeutic discovery is applicable to rare tumors and supports the advancement of brigatinib, as a multi-TKI into clinical trials for the treatment of NF2-associated meningiomas and schwannomas [71].

Drawing on previous research that designated phosphoinositide-3 kinase (PI3K) as a viable target [72], screens were conducted for PI3K pathway inhibitors to assess their effectiveness in diminishing the viability of human schwannoma cells [73]. The principal compound, CUDC907, a dual histone deacetylase (HDAC)/PI3K inhibitor, was further examined for its influence on isolated and nerve-grafted schwannoma model cells, as well as primary VS cells. CUDC907 reduced tumor growth rate by 44% in a 14-day treatment protocol, altered phospho-target levels, and lowered YAP levels. In five primary VS samples, CUDC907 lessened viability, triggered caspase-3/7 cleavage, and decreased YAP levels. Its effectiveness was associated with baseline phospho-HDAC2 levels. CUDC907 exhibits cytotoxic properties in NF2 schwannoma models and primary VS cells, indicating it as a potential candidate for clinical trials in NF2-related schwannomatosis [73].

Based on a comparative analysis of metabolic characteristics between *Nf2*-deficient and wild-type (WT) cells [74], the metabolic profiles of WT and *Nf2*-deficient MEFs, and *Nf2*-WT and *Nf2*-deficient murine Schwann cells were characterized, opening up a novel therapeutic approach for NF2 syndrome [75]. It was discovered that the most significant metabolic alteration was a notable elevation in fatty acid levels. Further exploration revealed that fatty acid synthase (FASN) inhibitors (Cerulenin, C75, and GSK2194069) selectively eradicated *Nf2*-deficient MEFs, schwannoma, and meningioma cells, confirming that these cells were also responsive to anti-Fasn siRNAs. Treatment with FASN inhibitors curtailed tumor growth in mouse xenograft studies. Additionally, re-introducing Merlin in the *NF2*-deficient schwannoma cells made them resistant to the FASN inhibitors, verifying their potential as effective agents for neurofibromatosis II pharmacotherapy [75].

## 7. Ongoing Clinical Trials for NF2 Syndrome

In addition to these experimental studies on treatment, extensive clinical studies have been performed through targeting vascular endothelial growth factor (VEGF) signaling pathway. Bevacizumab, commonly known by its brand name Avastin, is a monoclonal antibody that targets and inhibits VEGF. VEGF is a signaling protein that promotes the growth of new blood vessels (angiogenesis). This process is critical in both normal physiological contexts, such as wound healing and the menstrual cycle, and in pathological conditions, particularly cancer [76]. VEGF is produced by cells under hypoxic (low-oxygen) conditions. Tumor cells often experience hypoxia, leading to the production of VEGF. VEGF primarily binds to VEGF receptors (VEGFR) on the surface of endothelial cells (cells lining blood vessels) [77]. Bevacizumab works by binding to VEGF and preventing it from interacting with its receptors on the surface of endothelial cells. This inhibition blocks the angiogenesis signaling pathway, which reduces tumor blood supply by inhibiting the growth of new blood vessels and normalizing tumor vasculature. Bevacizumab is used to treat various cancers, including colorectal cancer, non-small-cell lung cancer, renal cell carcinoma, glioblastoma, and ovarian cancer [78].

Given its mechanism of action, bevacizumab has been explored as a treatment for schwannomas, especially in the context of NF2. Schwannomas are highly vascular tumors, and inhibiting angiogenesis can potentially reduce their growth and vascularity. The VEGF pathway is central to angiogenesis, as previously described. In the context of NF2-related schwannomatosis, the overexpression of VEGF by schwannoma cells leads to increased angiogenesis, which supports tumor growth and survival [79,80]. By targeting this pathway, bevacizumab can effectively inhibit the growth of these tumors. Therefore, Bevacizumab, as a VEGF inhibitor, has a potential to decrease the blood supply to the schwannomas by preventing VEGF from binding to its receptors on endothelial cells. This may reduce tumor growth and limit the expansion of existing tumors, and reduce the pressure exerted by tumors on adjacent nerves and tissues, potentially alleviating symptoms such as hearing loss and tinnitus. A study by Plotkin et al. (2009) demonstrated that bevacizumab could reduce the size of vestibular schwannomas in NF2 patients and improve hearing in some cases [81,82,83]. However, there are several potential issues with bevacizumab treatment. It necessitates regular parenteral administration and has been linked to side effects and drug resistance [84,85].

As a similar approach with VEGFR instead VEGF, Axitinib was used for the treatment of NF2-related schwannomatosis. It is a tyrosine kinase inhibitor (TKI) that targets multiple receptors involved in angiogenesis, including VEGFR-1, VEGFR-2, and VEGFR-3. By inhibiting these receptors, axitinib can reduce the formation of new blood vessels (angiogenesis), which is crucial for tumor growth and survival [86,87]. However, all patients with axitinib experienced drug-related toxicities, and it was more toxic and less effective than bevacizumab; for these reasons, further investigations were discouraged [88].

As continuing efforts to develop new and efficient therapeutics, there are still several ongoing clinical trials for NF2 syndrome (Table 1). Selumetinib is a kinase inhibitor, more specifically a selective inhibitor of the enzyme mitogen-activated protein kinase (MAPK or MEK) subtypes 1 and 2. These enzymes are part of the MAPK/ERK pathway, which regulates cell proliferation (i.e., growth and division) and is overly active in many types of cancer. Selumetinib was discovered by Array BioPharma and was licensed to AstraZeneca. It has been investigated for the treatment of various types of cancer, such as non-small-cell lung cancer (NSCLC) and thyroid cancer [89]. On 10 April 2020, Selumetinib was approved by the FDA for the treatment of pediatric patients 2 years of age and older with neurofibromatosis type 1 who have symptomatic, inoperable plexiform neurofibromas. In the phase 2 trial, most children with neurofibromatosis type 1 and inoperable plexiform neurofibromas had durable tumor shrinkage and clinical benefit from selumetinib [90] (NCT01362803). To learn more about the effects that the selumetinib has on participants with NF2-related schwannomatosis (NF2) related tumor, Cincinnati Children’s Hospital launched the phase 2 clinical trial of Selumetinib (NCT03095248) on June 2020 and the study is scheduled to be complete by June 2025.

The PI3K/AKT pathway is activated in human vestibular schwannoma (VS) and meningiomas. Since it is a convergence point for many growth stimuli, and it controls downstream cellular processes such as cell survival, cell proliferation, insulin response, stress response, and differentiation, the PI3K/AKT pathway is an attractive therapeutic target for VS. AR-42 is an orally bioavailable pan-histone deacetylase inhibitor (HDACi) with tumor inhibitory effects on schwannomas and meningiomas in vitro and in vivo. AR-42, also known as REC-2282 or OSU-HDAC42, was formerly licensed by Arno Therapeutics (Parsippany, NJ, USA) and now is licensed by Recursion Therapeutics (Salt Lake City, UT, USA). AR-42 was under development for the treatment of relapsed multiple myeloma, chronic lymphocytic leukemia (CLL), Hodgkin lymphoma, non-Hodgkin lymphoma, acute myeloid leukemia, myelodysplastic syndrome, and bladder cancer. AR-42 induced cell-cycle arrest at G2/M, triggered apoptosis, and decreased p-AKT levels in both VS and meningioma cells. In vivo, AR-42 inhibited the growth of schwannoma xenografts, induced apoptosis, and decreased AKT activation. Additionally, AR-42 markedly diminished meningioma tumor volumes in a Ben-Men-1 xenograft mouse model and tumors did not return when followed 6 months post treatment [91,92,93]. Collectively, these results demonstrate a high potency of AR-42 against NF2-associated VS and meningiomas. Under the name of “Efficacy and Safety of REC-2282 in Patients with Progressive Neurofibromatosis Type 2 (NF2) Mutated Meningiomas”, a phase 2/3 clinical trial (NCT05130866) began in June 2022 and is supposed to be complete by July 2027.

Crizotinib, sold under the brand name Xalkori among others, is an anti-cancer medication used for the treatment of non-small-cell lung carcinoma (NSCLC). It acts as an ALK (anaplastic lymphoma kinase) and ROS1 (c-ros oncogene 1) inhibitor. Crizotinib has an aminopyridine structure, and functions as a protein kinase inhibitor by competitive binding within the ATP-binding pocket of target kinases. Crizotinib is indicated for the treatment of metastatic non-small-cell lung cancer (NSCLC) or relapsed or refractory, systemic anaplastic large-cell lymphoma (ALCL) that is ALK-positive. It is also indicated for the treatment of unresectable, recurrent, or refractory inflammatory anaplastic lymphoma kinase (ALK)-positive myofibroblastic tumors (IMT) [94,95,96]. In a phase 2 clinical trial (NCT04283669), subjects with NF2-related schwannomatosis (NF2) and progressive vestibular schwannoma (VS) have been treated with crizotinib administered orally since February 2020. The trial is supposed to be complete by December 2025.

Several clinicians and the Children’s Tumor Foundation (CTF) have developed an adaptive platform-basket trial to screen multiple drugs against any type of progressive NF2-related tumor, called INTUITT-NF2, standing for ‘Innovative Trial for Understanding the Impact of Targeted Therapies in NF2’. This multi-arm phase II platform-basket screening study was designed to test multiple experimental therapies simultaneously in patients with NF2-related schwannomatosis with associated progressive tumors of vestibular schwannomas (VSs), non-vestibular schwannomas (non-VSs), meningiomas, and ependymomas. Leading the INTUITT-NF2 trial are Scott Plotkin, MD, PhD, of Massachusetts General Hospital and Jaishri Blakely, MD, of Johns Hopkins University. The foundation of this new trial is to rapidly and efficiently screen multiple therapies simultaneously so as to enable faster approval studies that have the highest indication for success. The U.S. Food and Drug Administration (FDA) recently approved the initiation of this phase 2 trial (NCT04374305); it began in June 2020 and is supposed to be complete by December 2030.

Recently, PRG S&Tech (Busan, Republic of Korea) completed a preclinical study using PRG-N-01, building on research from Park and his colleagues [24,25]. Upon verifying the outcomes of PRG-N-01’s preclinical efficacy evaluations, it was noted that there was no toxicity in normal cells, it did not disrupt the normal TGFβ signaling pathway, and it specifically targeted the protein–protein interaction (PPI) between TβR1 and RKIP proteins. Moreover, a decrease in the size of tumors in the vestibular nerve ganglia was noted in the PRG-N-01 25 mg/kg treatment group. Consequently, selective anti-tumor effectiveness is anticipated for tumors arising from NF2 syndrome in this clinical trial. This could facilitate the development of novel treatment alternatives for NF2 syndrome patients lacking other therapeutic options. As this clinical trial represents the first human study of PRG-N-01, participation will be limited to adult NF2 syndrome patients with progressive and/or symptomatic conditions who are not candidates for surgery. Additionally, the TGFβ signaling pathway is crucial for cardiovascular development, and its inhibition could potentially result in cardiac toxicity. In the preclinical 4-week repeat-dose toxicity studies of PRG-N-01, no cardiac toxicity related to the drug’s mechanism of action was detected. The TGFβ inhibitors galunisertib and vactosertib displayed no cardiovascular-related adverse events in their phase 1 clinical trials [97,98]. Nevertheless, vigilant monitoring from a safety standpoint is required when considering treatment, with a critical focus on cardiovascular health. The Ministry of Food and Drug Safety (MFDS) of the Republic of Korea recently approved the commencement of this phase 1/2 trial for PRG-N-01 (KCT0009520), which started in May 2024 and is expected to conclude by December 2028.

## 8. Concluding Remarks

In this review, we have navigated through the intricate landscape of NF2-related schwannomatosis (NF2) syndrome, a disorder characterized by the formation of benign yet debilitating tumors within the nervous system. Our exploration reveals that NF2 syndrome represents a paradigm of the critical interplay between genetic mutations and their phenotypic manifestations in cellular signaling pathways and protein–protein interactions (PPIs). The loss of function of the Merlin protein, a pivotal tumor suppressor encoded by the *NF2* gene, underscores the molecular pathology of the disease. These molecular insights not only enhance our understanding of tumor biology associated with NF2 syndrome but also underscore the potential therapeutic avenues that target these aberrant pathways. Multiple efforts to understand the causes of Nf2 syndrome and develop treatments, and ongoing clinical studies have been described. However, this short review does not include contributions from all researchers. Preclinical studies introduced in this review were mainly selected for being related to the ongoing clinical trials.

Recent advances in targeted therapies and the encouraging results from ongoing clinical trials offer a beacon of hope for individuals afflicted by this challenging condition. The strategic inhibition of specific molecular targets, such as the MEK inhibitors and novel approaches targeting the PI3K/AKT pathway, have shown promising results in curbing tumor growth and alleviating symptoms associated with NF2 syndrome. One of the interesting developments in clinicals trial is PRG-N-01, an innovative and a first-in-human treatment emerging from comprehensive studies focusing on the PPI critical in NF2 pathology. PRG-N-01 might be offering a promising therapeutic approach without interfering with the canonical TGFβ signaling pathway. Another outstanding clinical trial is REC-2282, targeting histone deacetylase (HDAC) with tumor inhibitory effects on schwannomas and meningiomas in vitro and in vivo, formerly under development for treatments of myeloma, leukemia, lymphoma, etc. This precision in targeting could lead to higher efficacy and lower side effects, representing a significant advancement in NF2 therapy. As research continues to unravel the molecular intricacies of NF2 syndrome, it paves the way for a new era of precision medicine where targeted therapies can be tailored to combat the specific genetic and molecular aberrations present in each patient. This not only promises to enhance the efficacy of treatment strategies but also minimizes adverse effects, leading to improved quality of life for those affected.

In conclusion, while the journey towards a cure for NF2 syndrome is ongoing, the synthesis of genetic insights and innovative therapeutic strategies illuminates the path forward. Through collaborative research efforts and clinical advances, we edge closer to transforming the management of NF2 syndrome, turning what was once a debilitating genetic condition into a manageable aspect of life for many.

## Figures and Tables

**Table 1 ijms-25-06558-t001:** The list of the ongoing clinical trials for NF2 syndrome.

NCT ^1^ (or KCT ^2^) Numbers	Treatment (Inhibitors)	Stage (Site)	Estimated Enrollment (Persons)	Study Type	Study Start (Year-Month) /Primary Completion (Estimated) /Study Completion (Estimated)
NCT03095248	Selumetinib (MEK inhibitor)	Phase II (USA)	34	Interventional	2017-05 /2024-06 /2025-06
NCT05130866	REC-2282 (HDAC inhibitor)	Phase II/III (USA)	92	Interventional	2022-06 /2027-01 /2027-07
NCT04283669	Crizotinib (ALK and ROS1 inhibitor)	Phase II (USA)	19	Interventional	2020-02 /2025-08 /2025-12
NCT04374305	Brigatinib (ALK and EGFR inhibitor) Neratinib (HER tyrosine kinase ^3^ inhibitor)	Phase II (USA)	100	Interventional	2020-06 /2029-12 /2030-12
KCT0009520	PRG-N-01 (TβR1/RKIP PPI inhibitors)	Phase I/II (Republic of Korea)	I: 36 II: 30	Interventional	2024-05 /2028-06 /2028-12

^1^ NCT: National Clinical Trial. ^2^ KCT: Korea Clinical Trial. ^3^ HER tyrosine kinase: pan-human epidermal growth factor receptor tyrosine kinase.

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
