# Peer review of "NF2-Related Schwannomatosis (NF2): Molecular Insights and Therapeutic Avenues"

_ijms, 2024, doi:10.3390/ijms25126558_

Round 1

Reviewer 1 Report

Comments and Suggestions for Authors

The manuscript is a good review of NF2-related schwannomatosis and the needs for molecular directed treatments, but a bit confusing in its scope. It could use some simplification of parts and should decide whether it will discuss all potential treatments or just those associated with 2 pathways.

To start, sometimes the manuscript makes contradictory statements. For instance line 52 states surgery is an effective treatment for pain, but later on mentions how surgery can cause risks. So perhaps it's better to just focus on surgery in one section so it doesn't sound contradictory.

The section titled "treatment for NF2" is really pre-clinical research on possible treatments. Current title is misleading. Also, it's unclear why the manuscript is meant to focus on YAP and RKIP but the pre-clinical section is lengthy and a bit disorderly, and not really clear why these specific studies were selected (out of so many for NF-2 related pathways inhibition preclinical studies out there). Why not just focus on therapeutic possibilities for YAP & RKIP only? Just clearly state that this is a manuscript specifically to discuss these 2 pathways as treatment potentials for NF2, rather than stating this is a comprehensive review which it is not.

It is reasonable, though, that all treatments currently in clinical trials are discussed, as there aren't that many and perhaps discuss how treatments focusing on the 2 pathways discussed here can be brought to trials.

Comments on the Quality of English Language

sentences can be simplified but otherwise no major editing needed.

Author Response

Dear Reviewer, 

We appreciate your time and effort to make our manuscript more competitive. 

Please review the revised manuscript based on the reviewers' comments.

Thanks

Sincerely,

BJP

Reviewer 2 Report

Comments and Suggestions for Authors

The authors have undertaken a comprehensive review of  ‘Molecular Insights and Therapeutic Avenues’ In NF2. The clinical part is in need of some work and authors must use the current nomenclature. It is impossible to have a review of this sort and completely ignore the extensive work on bevacizumab and the VEGf pathway.

Specific points

1.       Title and throughout ‘Neurofibromatosis Type 2 (NF2): Molecular Insights and Therapeutic Avenues’ -The correct name is now NF2-related schwannomatosis. Please correct throughout

2.       ‘Neurofibromatosis type 2 (NF2) syndrome is a rare genetic disorder primarily char- 28 acterized by the development of benign tumors in the central nervous system, including 29 bilateral vestibular schwannomas, brain, spinal tumors, and peripheral nerve tumors.’ -How rare? What is the birth incidence and prevalence from a source reference please

3.       ‘Adults typically present with hearing loss and balance disturbance, and children with ocular, dermatological, and neurological signs.’ -A clinical reference or two please

4.       ‘The abnormal gene can be inherited from either of the parents, and the risk of passing the gene to offspring from a parent is 50%.’ -This is only the case for heterozygotes. As half of de novo cases are mosaic for half the risk is less than 50%. Please address

5.       ‘NF2 syndrome is caused by (mutations) in the NF2 35 gene located in the long arm of chromosome number 22 (22q12.2).’ –‘mutations’ should be altered to germline or mosaic pathogenic variants

6.       ‘. NF1 is characterized by multiple café-au-lait spots and neurofibromas on the skin, while NF2 syndrome is (more) associated with bilateral vestibular schwannomas [18].’ -why ‘more’ -this suggests that café au lait and neurofibromas are still part of NF2. Although CaL are probably more common <2% have 6 or more and true neurofibromas are NOT a feature of NF2. Please rephrase

7.       ‘The NF2 gene was identified in the late 1980s and early 1990s’ -No it was identified in 1993. Previous research from 1986 suggested the location on 22q.

8.       ‘Interestingly, research has shown that Merlin interacts with TGFβ receptor 2 (TβR2)..’ -Several sentences start with ‘interestingly, please find an alternative word for some of these

9.       ‘Numerous attempts have been executed to effectively (manage) NF2 syndrome using mouse models and cell lines derived from human patients’ -manage is not the right word here. Perhaps develop treatment strategies

10.   ‘Recently, Selumetinib was approved on (Arpil) 10, 2020 by FDA for the treatment of pediatric patients 2 years of age and older with neu-rofibromatosis type 1 who have symptomatic,’ -spelling of April

11.   ‘To learn more about the effects that the selumetinib has on participants with neurofibromatosis type II (NF2) related tumor’ -please refer to point 1. Please use the new nomenclature. There is no need to reintroduce NF2 here by its full name

12.   Table ‘NCT051308 66 REC-2282 (HDAC inhibitor) Phase II/III (USA)’ -This trial is not just in USA

13.   The authors have  missed a trial of axitinib from New York

14.   The authors cannot have a review on therapeutic avenues without discussing VEGf and bevacizumab

Comments on the Quality of English Language

OK

Author Response

(The authors gave the same response as above.)

Round 2

Reviewer 2 Report

Comments and Suggestions for Authors

The authors have improved the manuscript especially with the section on bevacizumab, but there are still some issues with referencing

 Specific issues

1.  ‘Neurofibromatosis type 2 (NF2) syndrome is a rare genetic disorder primarily characterized by the development of benign tumors in the central nervous system, including bilateral vestibular schwannomas, brain, spinal tumors, and peripheral nerve tumors.’ -How rare? What is the birth incidence and prevalence from a source reference please

è  As you suggested, the following sentences “The estimated prevalence of NF2-related schwannomatosis is 1:50,000 with a birth incidence of 1:28,000 (Visit the following web link from NIH, https://www.ncbi.nlm.nih.gov/books/NBK1201/#:~:text=Prevalence,birth%20incidence%20of%201%3A28%2C000).” were added in the introduction.

è  This is NOT a primary source reference. Please cite an original epidemiological study for this figure

2. ‘NF2 is passed down in an autosomal dominant fash- 37 ion. Around half of the individuals diagnosed with NF2 have inherited it from a parent 38 who also has the condition. The other half develop NF2 due to a new (de novo) mutation 39 in the NF2 gene. Between 25% and 50% of those with a new NF2 mutation exhibit somatic 40 mosaicism for the mutation. If a parent is shown to have mosaic NF2, it rules out the 41 possibility that they have the inherited form of the disorder. Each child of a person with 42 NF2 has a potential 50% chance of inheriting the mutation: children of someone with a 43 germline mutation have a 50% risk, while children of someone with mosaic NF2 have a 44 lower than 50% risk. Once the NF2 mutation is identified in a family, genetic testing can 45 Citation: Kim, B.-H.; Chung, Y.-H.; Woo, T.-G.; Kang, S.-m.; Park, S.; Kim, M.; Park, B.-J. NF2-related schwannomatosis (NF2): Molecular Insights and Therapeutic Avenues. Int. J. Mol. Sci. 2024, 23, x. https://doi.org/10.3390/xxxxx Academic Editor(s): Received: date Revised: date Accepted: date Published: date Copyright: © 2024 by the authors. Submitted for possible open access publication under the terms and con[1]ditions of the Creative Commons At[1]tribution (CC BY) license (https://cre[1]ativecommons.org/licenses/by/4.0/). Int. J. Mol. Sci. 2022, 23, x FOR PEER REVIEW 2 of 16 be done before birth or during the early stages of pregnancy.’ -Please add a reference for this. The review above would be sufficient.

Author Response

Dear Reviewer 2, 

We appreciate your time and effort to make our manuscript better.

Please look into the attached file for the reply to the comments. 

Sincerely,

BJP
